# On the Degree of Plastic Strain during Laser Shock Peening of Ti-6Al-4V

**DOI:** 10.3390/ma16155365

**Published:** 2023-07-30

**Authors:** Sergey Mironov, Maxim Ozerov, Alexander Kalinenko, Ivan Zuiko, Nikita Stepanov, Oleg Plekhov, Gennady Salishchev, Lee Semiatin, Sergey Zherebtsov

**Affiliations:** 1Institute of Materials Science and Innovative Technologies, Belgorod National Research University, 308015 Belgorod, Russia; ozerov@bsu.edu.ru (M.O.); kalinenko@bsu.edu.ru (A.K.); zuiko@bsu.edu.ru (I.Z.); stepanov@bsu.edu.ru (N.S.); salishchev_g@bsu.edu.ru (G.S.); zherebtsov@bsu.edu.ru (S.Z.); 2World-Class Research Center “Advanced Digital Technologies”, State Marine Technical University, 198095 Saint Petersburg, Russia; 3Institute of Continuous Media Mechanics, Ural Branch of Russian Academy of Science, 614013 Perm, Russia; poa@icmm.ru; 4MRL Materials Resources LLC, Xenia Township, OH 45385, USA; slsemiatin@gmail.com

**Keywords:** Ti-6Al-4V, laser shock peening (LSP), microstructure, electron backscatter diffraction (EBSD)

## Abstract

Laser shock peening (LSP) is an innovative technique that is used to enhance the fatigue strength of structural materials via the generation of significant residual stress. The present work was undertaken to evaluate the degree of plastic strain introduced during LSP and thus improve the fundamental understanding of the LSP process. To this end, electron backscatter diffraction (EBSD) and nano-hardness measurements were performed to examine the microstructural response of laser-shock-peened Ti-6Al-4V alloy. Only minor changes in both the shape of α grains/particles and hardness were found. Accordingly, it was concluded that the laser-shock-peened material only experienced a small plastic strain. This surprising result was attributed to a relatively high rate of strain hardening of Ti-6Al-4V during LSP.

## 1. Introduction

Due to its excellent combination of low density and high strength, Ti-6Al-4V is widely used in the aerospace industry for the manufacture of the fan and compressor parts of jet engines. The cyclic nature of loading, which is typical for such applications, also imposes strict requirements for fatigue resistance. To enhance this property, Ti-6Al-4V products are, therefore, given a surface treatment prior to being put into service. These operations include laser shock peening (LSP), which is a relatively recent innovation. LSP comprises high-energy laser pulsing of the metal surface, which is typically applied to generate residual compressive stresses. The key advantage of LSP is its ability to produce a relatively thick (mm-scale) residual stress zone, as compared to conventional shot peening, without compromising surface quality. The beneficial influence of LSP on fatigue endurance has been reported in a number of recent works; e.g., [1,2,3,4,5,6,7,8,9,10,11,12,13,14]. A comprehensive review of recent achievements in this field has been provided by Wang et al. [2].

In particular, an approximately 20 to 40 percent increase in fatigue strength of laser-shock-peened samples has been reported by Jin et al. [3] and Zhang et al. [4], respectively. In terms of fatigue life, the enhancement effect has been found to range from ~140 pct. [5] to ~300 pct. [6]. The beneficial influence of LSP may be further amplified using a double-sided treatment [7]. By this means, a ~170-pct. increase in fatigue endurance limit has been found in the work of Yang et al. [8]. Moreover, the combination of LSP with conventional shot peening has been shown to result in a nine-fold increase in the number of cycles-to-failure [9]. As shown by Spanrad et al. [10], Lin et al. [11], and Zabeen et al. [12], the superior fatigue properties of laser-shock-peened Ti-6Al-4V are associated with a retardation of both the nucleation and propagation of fatigue cracks by residual compressive stresses generated in near-surface areas. Furthermore, the distribution of residual stress (and thus fatigue resistance) was found to be highly sensitive to the particular LSP pattern/grid applied during the treatment [13]. A pattern consisting of multiple concentric rings of LSP spots was found to produce the most favorable distribution of residual stresses. Given the numerous benefits of LSP treatment, it has thus been recommended for use as a standard practice in damage-tolerant designs [14].

The superior efficiency of LSP treatment is likely associated with local microstructure and/or substructure changes in the processed material. Hence, microstructure studies in this area are of significant practical interest. Furthermore, given the extreme processing conditions associated with LSP, such as GPa-level imposed stresses, ultra-high strain rates (>10^6^ s^−1^), and ultrashort durations (~10^−8^ s), such investigations may broaden our fundamental understanding of dynamic material behavior.

A review of the scientific literature on the LSP of Ti-6Al-4V reveals a somewhat contradictory range of experimental observations. For example, the development of a nanocrystalline grain structure in the surface layer of laser-shock-peened material has been reported in several works [3,15,16]. This effect has been attributed to the development of dynamic recrystallization [3,15,16,17] enhanced by adiabatic heating [15,16]. On the other hand, only minor changes in grain shape and size have been observed in a number of other studies [18,19,20]. Furthermore, extensive mechanical twinning has often been noted [3,15,16,17,21,22,23,24,25]. However, only subtle twinning was detected by Laine et al. [18]. Finally, the dislocation density has been typically reported to be very high [17,21,22,23,25,26]. Nevertheless, quantitative analysis has shown dislocation densities of only ~10^14^ m^−2^ [3] or even ~10^13^ m^−2^ [24]. For the microstructure of Ti-6Al-4V in some heat-treatment conditions (e.g., mill-annealed), such densities are comparable with those in the material prior to LSP. In addition, it is interesting to note that planar dislocation structures have been observed in some works [17,18,19]; i.e., structures that are typically attributable to the early stages of plastic flow. A number of microstructural observations have suggested that LSP results in large (or even severe) plastic deformation [3,15,16,17,21,22,23,24], while others appear to indicate a small LSP-induced strain [18,19,20].

In an attempt to shed more light on the structural response of Ti-6Al-4V to LSP, the present work was undertaken with the purpose of evaluating the degree of the LSP-induced plastic strain (at least qualitatively). According to recent numerical simulations, this strain could be as low as ~0.01 [27]. Thus, the present work focused on the careful examination of LSP-induced microstructure to confirm or disprove this result. To the best of the authors’ knowledge, the present work is one of the first systematic studies in this field. It is believed that the elucidation of this issue should contribute to our fundamental understanding of material behavior under the extreme conditions of LSP.

## 2. Materials and Methods

The material used in the present investigation was commercially produced Ti-6Al-4V alloy in mill-annealed condition. The nominal impurity content includes 0.25 wt. pct. (max) of iron and 0.2 wt. pct. (max) of oxygen. Standard mill-processing involves hot rolling at 900 °C followed by annealing at 700 °C and final air cooling. This processing route is commonly used for turbine-blade applications and thus benefits from surface treatment via the LSP technique. The microstructure of the program material was dominated by fine, globular, primary α phase with a minor fraction of retained β phase (Figure 1).

To produce the LSP-induced microstructures, the following procedure was adopted. A series of workpieces measuring 30 (length) × 14 (width) × 6 (thickness) mm^3^ was machined from the original hot-rolled sheet (Appendix A). The front surface of each workpiece was mechanically polished to a mirror finish, cleaned in water, and degreased in ethanol. Then, the central location of the front surface was subjected to one, three, or five successive laser pulses in order to examine the possible sensitivity of microstructural changes to the number of LSP passes. Importantly, no systematic peening of the workpieces was applied; rather, all laser pulses were focused on the same (central) spot.

In all cases, LSP was conducted using a Q-switched Nd:YAG laser with a wavelength of 1064 nm, which was operated at 10 Hz and had a pulse duration (full width at half maximum) of 20 ns. To provide comparatively severe deformation conditions during LSP, the laser pulse energy and the spot size were set at 5 J and 1 × 1 mm^2^, thus giving a power density of 25 GW/cm^2^. To protect the material surface from ablation effects, 0.1-mm-thick steel foil was applied as a sacrificial layer. Running water was used as the transparent confining medium. It is well established that the propagation of a laser pulse across the interface between two media with distinctly different densities (i.e., the water layer and the sacrificial steel layer in the present study) should result in a plasma spot, which, in turn, provides the LSP effect.

The LSP-induced microstructures were characterized using the electron backscatter diffraction (EBSD) technique. This choice was dictated by the extremely localized character of microstructure changes during LSP. According to the scientific literature [18,19], the plastically deformed zone developed during the LSP of Ti-6Al-4V may only be a few grains deep. Hence, the preparation of an appropriate microstructural sample for transmission-electron microscopy (TEM) or X-ray diffraction (XRD) analysis is technically challenging. An additional issue is the inhomogeneous distribution of the microstructure within the processed zone. Hence, statistical reliability is of particular importance for its examination. By contrast, the advanced capabilities of EBSD provide in-depth insight into the microstructure, thus enabling a more or less complete picture of underlying processes.

For microstructural observations, each peened sample was sectioned through its thickness and prepared using conventional metallographic techniques. The final surface finish was obtained by 24-h vibratory polishing with a colloidal-silica suspension. EBSD was performed with an FEI Quanta 600 scanning electron microscope equipped with a TSL OIM system and operated at an accelerating voltage of 20 kV. The total statistics of EBSD measurements are given in Table 1.

Two particular EBSD characteristics, viz., the image-quality (IQ) index and kernel average misorientation (KAM), were used for microstructural analysis. The IQ index quantifies the sharpness of the Kikuchi bands and thus serves as a metric for lattice defects. KAM is the average misorientation angle of a given pixel in an EBSD map relative to all of its neighbors with the proviso that misorientations above a certain threshold are excluded from consideration. Therefore, KAM reflects the local orientation spread and can be used as a measure of the excess density of dislocations of the same sign and local lattice distortion/curvature. In the present study, the KAM index was calculated for the nearest six neighbors using a threshold misorientation of 5 degrees. To minimize experimental error, KAM data were derived from EBSD maps obtained with the same scan step size of 0.2 µm.

To assist in the interpretation of microstructural changes, a nanohardness map was also measured. For this purpose, Berkovich nanohardness measurements were performed employing a Rtec SMT-5000 nanohardness tester by applying a 10-g load, a dwell time of 10 s, and a step size of 20 µm.

## 3. Results

Microstructural examination of the material subjected to a single LSP pass revealed no significant changes (Figure 1 vs. Figure 2). Hence, any such changes were likely relatively small and likely beyond the detection limit of EBSD. In the following two sections, therefore, attention was focused on microstructure evolution only during three or five LSP passes.

### 3.1. Three LSP Passes

#### 3.1.1. Surface Phenomena

The EBSD IQ map determined for the material subjected to three LSP passes (Figure 3a,b) revealed the formation of a subtle crater (of ~1 µm depth) on the pulsed surface. A change in surface topography after the LSP of Ti-6Al-4V has been previously reported [9,22]. Significantly, α grains/particles beneath the crater exhibited no clear evidence of compression (Figure 3a). Hence, it may be concluded that the crater likely originated from the partial ablation of surface material, which is often observed during high-energy laser pulsing.

In this context, the formation of a narrow dark-contrast layer at the crater surface is of interest (arrows in Figure 3b). High-resolution EBSD revealed no measurable variation in contrast (i.e., the presence of internal structure) within this layer, thus suggesting its amorphous nature. Indeed, the amorphization of surface material during high-energy laser pulsing has been occasionally reported [28,29]. Although the origin of this phenomenon is still not completely clear, a plausible explanation is the occurrence of the rapid solidification of ablated material [29].

It is also worth noting that the microstructure within the circled area in Figure 3b exhibited what appears to be either a Widmanstatten or a martensitic alpha microstructure. In titanium alloys, such microstructures are usually associated with high cooling rates; e.g., [30,31]. It is also interesting to note that the morphology of the Widmanstatten structure is somewhat similar to microstructural patterns sometimes ascribed to multiple twinning in several previous studies of laser-shock-peened Ti-6Al-4V (Appendix A).

From the above considerations, it is possible that the development of the nanocrystalline structure as well as extensive multiple twinning, which are sometimes reported in the scientific literature, e.g., [3,15,16,17], are actually artifacts associated with the rapid solidification of ablated material. On the other hand, it should be noted that the program material used in most prior efforts had a transformed-β microstructure [3,15,17], in contrast to the mill-annealed material employed in the present study. Hence, the possibility exists that differences in structural response to LSP are related to the initial microstructure.

#### 3.1.2. Subsurface Processes

The key microstructural characteristics of the material beneath the surface layer included mechanical twins (given the fine-grain nature of mechanical twins, particular care was taken to confirm the activation of the twinning mechanism, as exemplified in Appendix A.) and deformation bands in α phase (Figure 3a). On the other hand, no significant changes in either grain shape or grain size (Figure 3a) were found.

In agreement with previous work [19], the misorientations across the twin boundaries were typically close to $85^{o}<11\bar{2}0>$ (highlighted in blue in Figure 3a), thus suggesting the activation of $\{10\bar{1}2\}<\bar{1}011>$ twinning. However, twinning was a sporadic in nature.

Deformation bands appeared as a series of nearly parallel dark-contrast bands within α grains (Figure 3a). Compared to twinning, deformation bands were observed in most of the grains beneath the LSP crater (Figure 3a), and thus represented a typical phenomenon. The band traces were typically close to the traces of $\left\{11\bar{2}0\right\}$ prism planes, as indicated by the broken lines in Figure 3a. To obtain further insight into the origin of the bands, orientation gradients within α grains were measured (Figure 4). The local orientation spreads associated with the bands were usually below 2°, thus being *lower* than the angular resolution of EBSD. The deformation bands were thus interpreted as poorly developed dislocation boundaries, which were mainly produced by prism slip.

From the KAM map (Figure 4a), it was also noted that the largest orientation spreads (and thus the highest dislocation densities) were observed within the Widmanstatten structure (circled area) as well as near phase and grain boundaries. At the scale of α grains, however, the cumulative orientation gradients were typically below ~0.5°/µm (blue lines in Figure 4b–d), thus being relatively low.

### 3.2. Five LSP Passes

EBSD maps obtained after five LSP passes (Figure 5 and Figure 6) revealed a significant broadening of the deformation zone, as shown by a comparison of Figure 3a and Figure 6a. This effect was primarily due to the formation of three LSP craters on the material surface (Figure 6a) rather than one (Figure 3a). On the other hand, the depth of the deformation zone did not significantly change, still being ~50 µm (Figure 5a).

After five LSP passes, a significant degradation of IQ contrast was also found (Figure 3a and Figure 5b). This effect was obviously associated with an increase in dislocation density, and is discussed in greater detail in Section 4.1. Furthermore, a marked increase in $\left\{10\bar{1}2\right\}$ twinning was found (Figure 5b). The twinning was most pronounced in proximity to LSP craters, and thus led to significant grain refinement in these areas (Figure 4 and Figure 5b). Nevertheless, except for the twinning, the α grains/particles exhibited no significant change in morphology (Figure 5b). That is to say, no grain compression (or any other regular change in grain shape) was observed. As for the material subjected to three LSP passes, the substructure within the α grains after five passes was still dominated by poorly developed dislocation boundaries (Figure 5b). In some cases, several intersecting sets of the boundaries were found.

Compared to the results for three LSP passes, an increase in the orientation spread within α grains/particles was found for material subjected to five LSP passes (Figure 4a vs. Figure 6a). On the other hand, the magnitude of local orientation fluctuations (red lines in Figure 6b–d) was still close to the resolution limit of EBSD (i.e., 2°). Although poorly defined deformation-induced boundaries were observed, no evidence of extensive grain subdivision was found. Within the entire deformation zone, the highest orientation spread was associated with twinned areas as well as with phase and grain boundaries (Figure 6a).

Considering the GPa level of stress applied during LSP, the pulsed material was also examined for the presence of omega phase. However, no reliable evidence of this phase was found. Moreover, no sample-scale deformation/shear bands or signs of adiabatic heating were observed.

## 4. Discussion

### 4.1. Evaluation of the Degree of the LSP-Induced Strain

One of the most striking characteristics of the LSP-induced microstructure was the minimal (if any) change in the shape of the α grains/particles (it should be noted that no quantitative analysis of grain shape was applied in the present study, and the derived conclusion was solely based on a qualitative assessment of microstructures.) (Figure 3a and Figure 6b). Another important issue was the poorly developed dislocation substructure within the α phase; i.e., a characteristic usually attributable to relatively early stages of plastic flow. Both of these results provided evidence that the *plastic* deformation experienced during LSP was very small. To validate this result, the dislocation density within the deformation zone was estimated.

According to He et al. [32], the density of geometrically necessary dislocations ρ can be derived from EBSD measurements as ρ≈θ/xb, in which *θ* is the mean KAM angle, x is EBSD scan step size, and b is the Burgers vector. To minimize the measurement error, KAM data for all examined material conditions were derived from EBSD maps, which were obtained using a relatively coarse scan step size of 0.2 µm. Given the presumed prevalence of the prism slip in the α phase, the Burgers vector was taken to be 0.295 nm. For the β phase, the Burgers vector was taken to be 0.2837 nm.

From the calculations, it was found that the dislocation density in both phases was ~10^14^ m^−2^ (Figure 7c). These results were consistent with the literature data [3,24]. From the KAM maps, it was expected that dislocation density greatly varied within the laser-shock-peened material. Specifically, it was presumed to be highest within the solidified (Widmanstatten/martensitic) structure, as well as in twinned areas, but it was comparatively low in the α grains lying between the LSP craters. However, the *average* density of geometrically necessary dislocations was not excessively high.

It should be emphasized that EBSD cannot be used to quantify dislocations that produce no orientation spread (i.e., so-called *statistically-stored* dislocations). Hence, the above estimates were not conclusive in establishing the effect of LSP on the total dislocation density. Thus, to obtain insight into this issue, nanohardness measurements were conducted in the material subjected to five successive LSP passes (i.e., the presumably most heavily deformed LSP condition). The resulting nanohardness map, which included both the LSP-induced deformation region as well as the unaffected (surrounding) material, is shown in Figure 8. Inasmuch as the size of each nanohardness indent was comparable to the typical dimensions of α and β particles, the measurements were “contaminated” by unavoidable experimental scatter. Nevertheless, it was still clear that LSP provided no significant strengthening effect. In this context, it is important to emphasize that the laser-shock-peened material is typically characterized by significant residual stresses, which may be as high as ~800 MPa; e.g., [19]. Thus, it is unclear why such stresses would appear to exert no distinct influence on hardness.

The entire set of experimental results thus suggested that the magnitude of the LSP-induced plastic strain was *low.* This conclusion is consistent with numerical simulations of LSP [27], according to which the true plastic strains generated during LSP of Ti-6Al-4V were as low as 0.01. In turn, the comparability of these results is perhaps indicative of the feasibility of numerical approaches for an analysis of the behavior of Ti-6Al-4V under laser treatment, as highlighted by Jin et al. [33].

In the context of the above conclusion, it is worth noting that the LSP conditions in the present study were selected to provide severe deformation (i.e., a power density of 25 GW/cm^2^ and five successive LSP pulses to nearly the same location). Nevertheless, the generated plastic strain was found to be low. Therefore, it is likely that a typical LSP condition, which involves much fewer severe deformation conditions, should result in an even smaller strain. Hence, the conclusions derived in this work are likely applicable to the LSP of Ti-6Al-4V in general.

### 4.2. Magnitude of LSP-Induced Stress

As shown above, comparatively low plastic strain is likely an intrinsic characteristic of the LSP of Ti-6Al-4V. The possible origin of this phenomenon is considered in the following two sections.

One of the simplest explanations for the low LSP strain is the relatively high dynamic yield strength of Ti-6Al-4V. Indeed, this material exhibits a relatively high strength under static loading conditions (~1 GPa). Therefore, the possibility exists that the dynamic strength of this alloy at typical LSP strain rates (≥10^6^ s^−1^) may become comparable to the magnitude of the imposed shock stress. This hypothesis is considered in the present section.

First, it is important to emphasize that experimental measurements of dynamic strength (and strain) are limited by the highest strain rate achievable using the split Hopkinson bar tests; i.e., 10^4^ s^−1^. This is two orders *lower* than the typical LSP strain rate. Hence, the so-called Hugoniot elastic limit is typically used to evaluate the dynamic yield strength of materials during LSP. This measure characterizes the highest elastic stress in the direction of the shock wave propagation, often considered to be independent of strain rate. In Ti-6Al-4V, the Hugoniot elastic limit has been reported to be ~2.9 GPa; e.g., [34].

On the other hand, the peak stress generated during LSP is often evaluated using a simple one-dimension model of laser ablation in a confined environment, as proposed by Fabbro et al. [35] (from theoretical considerations, it was determined that thermal effects induced during LSP were mainly confined within the sacrificial layer and were small in the bulk of the Ti-6Al-4V workpiece; see details in the Appendix A “Supplementary material_Evaluation of thermal effect of LSP”; hence, thermal effects were neglected during the evaluation of the LSP stress). According to this approach, the propagation of a laser pulse through the interface between two media with distinctly different densities (i.e., the water layer and the sacrificial steel layer in the present study) results in a plasma spot with a size of
(1)$$L(\tau )=\int_{0}^{\tau }(u_{1}+u_{2})\partial \tau ,$$
in which $u_{1}$ and $u_{2}$ denote the extension rates of the plasma spot in water and steel, respectively, and τ is the duration of laser pulsing. This gives rise to a pressure of
(2)$$P_{i}=\rho_{i}D_{i}u_{i}=Z_{i}u_{i},$$
where $\rho_{i}$ is density, $D_{i}$ is the propagation rate of a longitudinal elastic wave, $Z_{i}$ is an acoustic impedance, and i is the index denoting the particular medium (i={1,2}). Hence,
(3)$$P(\tau )\frac{Z_{1}+Z_{2}}{Z_{1}Z_{2}}=P(\tau )Z=\frac{\partial L}{\partial \tau },$$
where $Z_{1}$ is the acoustic impedance of water (=0.15 × 10^6^ g/(cm^2^ × s)) and $Z_{2}$ is the acoustic impedance of steel (=4.5 × 10^6^ g/(cm^2^ × s)).

The density of the surface energy of the plasma spot can be expressed as
(4)$$J(\tau )=P(\tau )\frac{\partial L}{\partial \tau }+\frac{\partial \left(E(\tau )L(\tau )\right)}{\partial \tau },$$
in which E(τ) is the plasma internal energy.

The total energy of plasma can be defined as the sum of the thermal energy $E_{T}(\tau )$ and ionization energy $E_{I}(\tau )$:(5)$$E(\tau )=E_{T}(\tau )+E_{I}(\tau ),$$

The fraction of thermal energy can be defined as $E_{T}(\tau )=\alpha_{T}E(\tau )$, where $\alpha_{T}$ is often taken as 0.25 [36,37,38,39,40,41].

Considering the plasma as an ideal gas, its peak pressure should be directly related to the thermal energy, and thus can be expressed as
(6)$$P(\tau )=\frac{2}{3}E_{T}(\tau )=\frac{2}{3}\alpha_{T}E(\tau ),$$

Taking into account Equations (3) and (6), Equation (4) can be rearranged as
(7)$$ZJ\left(\tau \right)=ZP\left(\tau \right)\frac{\partial L\left(\tau \right)}{\partial \tau }+\frac{\partial \left(ZE\left(\tau \right)L\left(\tau \right)\right)}{\partial \left(\tau \right)}=\frac{\partial L}{\partial \tau }\frac{\partial L}{\partial \tau }+\frac{3}{2\alpha_{T}}\frac{\partial \left(\frac{\partial L\left(\tau \right)}{\partial \tau }L\left(\tau \right)\right)}{\partial \tau }=\;\frac{\partial L^{2}}{\partial \tau }+\frac{3}{2\alpha_{T}}\frac{\partial \left(L\left(\tau \right)\frac{\partial L}{\partial \tau }\right)}{\partial \tau }=\frac{\partial L^{2}}{\partial \tau }+\frac{3}{2\alpha_{T}}\left(\frac{\partial L^{2}}{\partial \tau }+L\left(\tau \right)\frac{\partial^{2}L\left(\tau \right)}{\partial \tau^{2}}\right),$$

Equation (7) can be numerically solved. Assuming that $J=const=J_{0}$, the characteristic size of the plasma spot can be approximated as
(8)$$L\left(\tau \right)=\sqrt{ZJ_{0}\frac{2\alpha_{T}}{3+2\alpha_{T}}}\tau ,$$

Hence, the peak stress generated by the shock wave can be found as
(9)$$P_{peak}=\frac{\frac{\partial L}{\partial \tau }}{Z}=\sqrt{\frac{J_{0}}{Z}\frac{2\alpha_{T}}{3+2\alpha_{T}}},$$

Taking the laser power density $J_{0}$ to be 25 GW/cm^2^, Equation (9) gives a peak imposed pressure of 7.2 GPa.

It is important to emphasize that the interaction of the laser beam with the water layer may also give rise to a plasma. It is opaque to the laser beam, and thus it may absorb a portion of the laser energy that exceeds a certain threshold. This situation has been considered in a number of works, which show that the threshold energy lies in the range of 8–10 GW/cm^2^ [42,43]. Accordingly, the magnitude of the pulse-induced pressure was predicted to saturate at ~4 GPa and shows only subtle changes with a further increase in laser power density above 8–10 GW/cm^2^ [42].

Despite the saturation of the LSP-generated pressure, its peak magnitude (~4 GPa) significantly exceeded the Hugoniot elastic limit of Ti-6Al-4V (~2.9 GPa). Hence, the relatively small plastic strain observed in the laser-shock-peened material was unlikely due to its high dynamic strength. Nevertheless, it is important to emphasize that both the shock stress and dynamic strength of Ti-6Al-4V were not experimentally measured in the present study but were derived from theoretical considerations or numerical simulations. Therefore, the initial hypothesis proposed in this section cannot be completely ruled out.

### 4.3. Strain Hardening Rate of LSP

Assuming that the above stress estimates are correct, the most plausible explanation for the low LSP strain appears to be a relatively high *strain hardening rate* during the propagation of the shock wave. Considering a linear relationship between stress and strain and assuming the accumulated strain to be ~0.1, the strain hardening rate in the present work should be as high as ~10 GPa. This magnitude is comparable to the strain hardening rate of ~3 to 5 GPa, which was recently measured for split Hopkinson-bar tests of Ti-6Al-4V at strain rates of ~10^3^ s^−1^ [44].

Such a high strain hardening rate is likely associated with the ultrashort duration of the LSP process (~10^−8^ s), which is perhaps consistent with an absence of thermal-assisted mechanisms of dislocation slip (e.g., dynamic recovery). Hence, plastic flow is likely solely governed by stress.

## 5. Conclusions

The present work was undertaken to fill the gap in the current scientific literature regarding the degree of plastic strain induced during LSP. It was believed that an elucidation of this issue would provide fundamental insight into the behavior of structural materials under the extreme conditions of LSP.

To achieve the above purpose, the structural response of Ti-6Al-4V to LSP was investigated using extensive EBSD and nanohardness measurements. Considering the minor changes in both grain shape and hardness, it can be concluded that laser-shock-peened material experiences only small plastic strains. This surprising result is attributed to the relatively high strain hardening rate during LSP and the absence of dynamic recovery.

## Figures and Tables

**Figure 1 materials-16-05365-f001:**
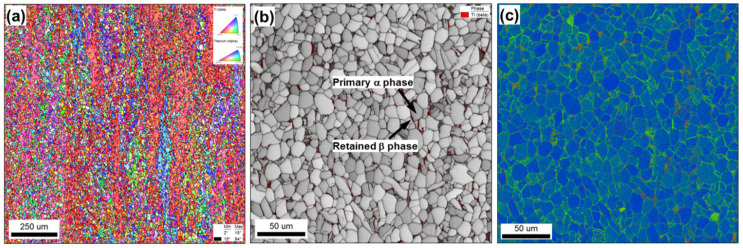
Microstructure of the initial material: (**a**) Low-resolution inverse pole figure map, (**b**) high-resolution image quality map with overlaid β phase (in red), and (**c**) high-resolution kernel average misorientation map.

**Figure 2 materials-16-05365-f002:**
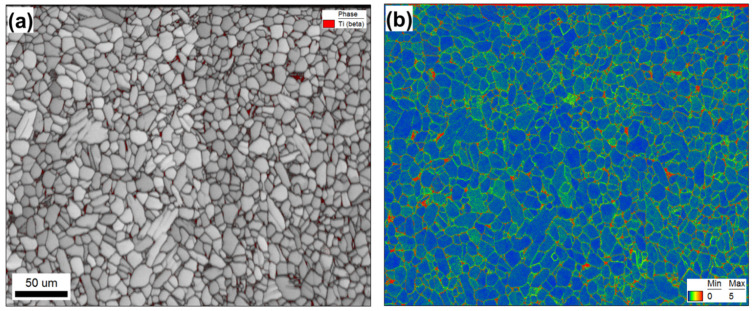
EBSD maps taken after a single LSP pass: (**a**) Image quality map with overlaid β phase (in red) and (**b**) kernel average misorientation (KAM) map. The laser-peened surface is at the top. In (**b**), the KAM color code is shown in the bottom right corner.

**Figure 3 materials-16-05365-f003:**
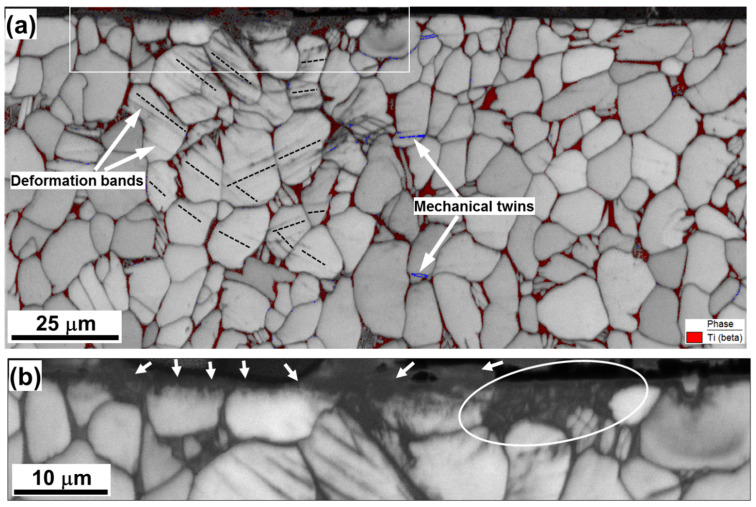
(**a**) Image quality map taken after three successive LSP pulses from the material beneath the LSP crater with the selected area shown at higher magnification in (**b**). The laser-peened surface is at the top. For illustrative purposes, β phase in (**a**) is highlighted with red; in α phase, blue lines indicate {10–12} twin boundaries, and black broken lines show {11–20} plane traces, which are closest to the orientation of deformation ands. In (**b**), arrows indicate the presumed amorphous layer, while the circled area shows a Widmanstatten (or martensitic) alpha microstructure.

**Figure 4 materials-16-05365-f004:**
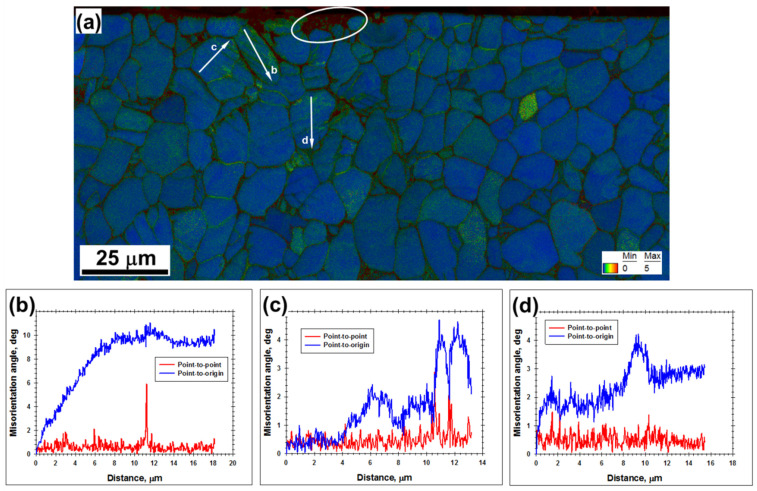
(**a**) Kernel average misorientation (KAM) map with overlaid IQ map taken after three successive LSP pulses from the material beneath the LSP crater, and (**b**–**d**), misorientation profiles measured along the arrows labeled in (**a**). The color code for KAM angles is shown in the bottom right corner of (**a**). The circled area in (**a**) shows the presumed Widmanstatten (or martensitic) alpha structure.

**Figure 5 materials-16-05365-f005:**
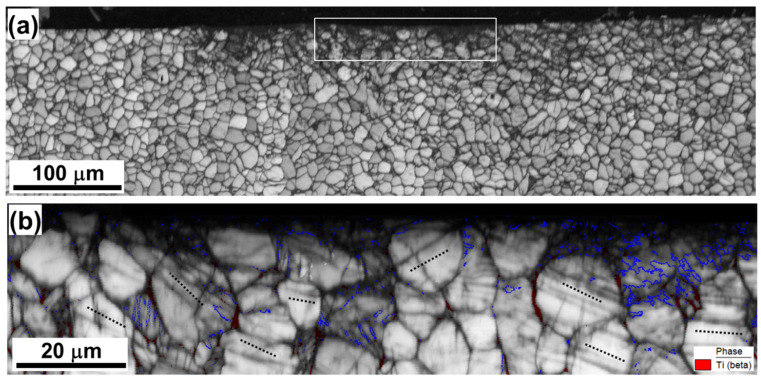
(**a**) Image quality map taken after five LSP passes from the material beneath the pulsed region with the selected area shown at a higher magnification in (**b**). The laser-peened surface is at the top. In (**b**), β phase is highlighted with red; in α phase, blue lines in (**b**) indicate {10–12} twin boundaries, and black broken lines show {11–20} plane traces, which are closest to the orientation of deformation bands.

**Figure 6 materials-16-05365-f006:**
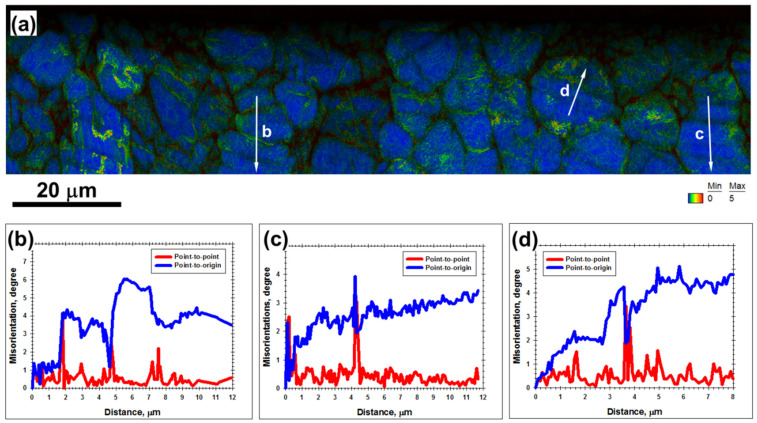
(**a**) Kernel average misorientation (KAM) map taken after five LSP passes from the material beneath the pulsed area, and (**b**–**d**), misorientation profiles measured along the arrows shown in (**a**). The color code for KAM angles is shown in the bottom right corner of (**a**).

**Figure 7 materials-16-05365-f007:**
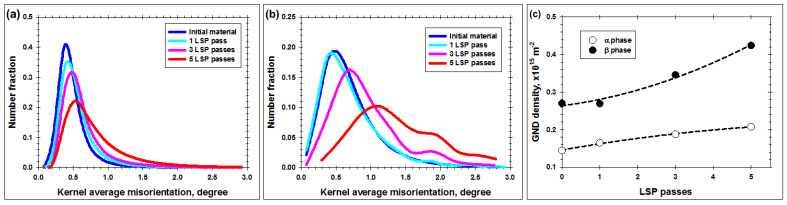
Effect of LSP passes on the distribution of kernel average misorientations in (**a**) α phase, (**b**) β phase, and (**c**) evolution of the density of geometrically necessary dislocations.

**Figure 8 materials-16-05365-f008:**
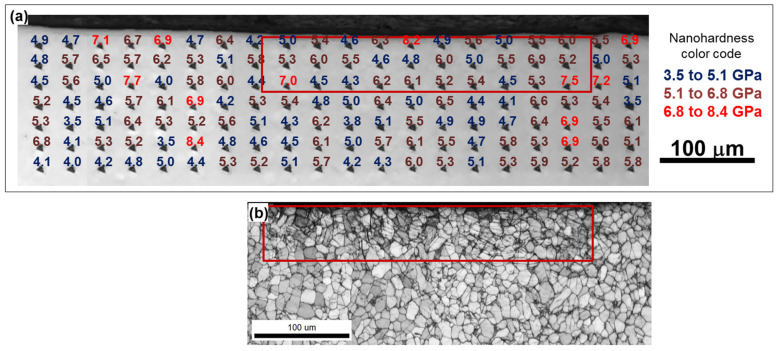
Effect of five LSP passes on hardness: (**a**) nanohardness map measured near the LSP-induced deformation zone and (**b**) EBSD image quality map taken from the same area before the hardness measurements. In (**a**), the numbers near hardness indents show the measured hardness magnitude in GPa. In both figures, the selected area approximates the LSP-induced deformation zone.

**Table 1 materials-16-05365-t001:** Statistics of EBSD measurements.

Material Condition	Scan Step Size, µm	Acquired Area, µm^2^	Number of Pixels	Average Confidence Index
Initial state	1	1300 × 1300	1,953,350	0.52
	0.2	500 × 500	7,218,944	0.61
1 LSP pass	1	3460 × 1000	3,998,033	0.34
	0.2	695 × 320	6,436,626	0.57
		348 × 287	2,880,695	0.51
		1384 × 300	11,996,693	0.49
		348 × 211	2,115,755	0.55
3 LSP passes	1	3458 × 1026	4,101,781	0.26
	0.2	694 × 207	4,141,389	0.48
		695 × 250	5,020,066	0.45
		695 × 350	7,023,986	0.47
	0.05	174 × 80	6,422,724	0.56
5 passes	1	3461 × 1798	7,189,536	0.37
	0.2	695 × 300	6,023,042	0.46
		630 × 300	5,459,817	0.50
		695 × 300	6,024,775	0.52
		695 × 250	5,020,066	0.58
	0.1	345 × 100	3,985,328	0.46

## Data Availability

Data available on request.

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
