# Peer review of "On the Degree of Plastic Strain during Laser Shock Peening of Ti-6Al-4V"

_materials, 2023, doi:10.3390/ma16155365_

Round 1

Reviewer 1 Report

This study deals with "On the degree of plastic strain during laser shock peening of 2 Ti-6Al-4V". The manuscript has enough innovation and provides valuable data. However, it is not well organized, the authors must check these comments:

1- Some sentences throughout the manuscript have grammatical errors. They need to be corrected in terms of grammar and structure. 

2-The abstract does not have a proper structure. It is necessary to rewrite it to summarize and clearly show the manuscript's purpose. Moreover, the last paragraph of the introduction also has the same problem.

3- At the end of the first paragraph of the introduction (Refs. 1-14), distribute them throughout the paragraph according to the meaning of the text or mention the most important ones. There is no need to refer so many references at once.

4- At the beginning of the materials and methods section, the impurity level of the consumed materials should be mentioned.

Overall, authors are strongly advised to make the text coherent. The text and the structure of the article are not coherent. This will be boring for the readers.

It must be rewritten according to the comments.

Author Response

Responses to the comments of Reviewer #1

 on the paper entitled “On the degree of plastic strain during laser shock peening of Ti-6Al-4V” (Manuscript ID: materials-2508720)

The authors would like to express their gratitude to Reviewer for his/her remarks. Below, we provided specific replies to the issues raised.

Note: Reviewer’s comments are highlighted with bold. 

This study deals with "On the degree of plastic strain during laser shock peening of 2 Ti-6Al-4V". The manuscript has enough innovation and provides valuable data. However, it is not well organized, the authors must check these comments:

1- Some sentences throughout the manuscript have grammatical errors. They need to be corrected in terms of grammar and structure. 

Authors’ response

According to the comment, the manuscript has been carefully checked with regard to English grammar usage.

2-The abstract does not have a proper structure. It is necessary to rewrite it to summarize and clearly show the manuscript's purpose. Moreover, the last paragraph of the introduction also has the same problem.

Authors’ response

According to the comment, the abstract and introduction sections of the manuscript have been revised in order to emphasize the purpose of the present study.

Abstract:

"…Laser shock peening (LSP) is an innovative technique that is used to enhance the fatigue strength of structural materials via the generation of significant residual stress. The present work was undertaken to evaluate the degree of plastic strain introduced during LSP and thus improve fundamental understanding of the LSP process."

Introduction (Page 2, Paragraphs 3 to 4):

"…A number of microstructural observations have suggested that LSP results in large (or even severe) plastic deformation [3, 15-17, 21-24] while others appear to indicate a small LSP-induced strain [18-20]. 

In an attempt to shed more light on the structural response of Ti-6Al-4V to LSP, the present work was undertaken with the purpose of evaluating the degree of the LSP-induced plastic strain (at least qualitatively)."

3- At the end of the first paragraph of the introduction (Refs. 1-14), distribute them throughout the paragraph according to the meaning of the text or mention the most important ones. There is no need to refer so many references at once.

Authors’ response

According to the comment, the key results of Refs. [1-14] have been detailed in the revised manuscript (Pages 1 to 2):

"…The beneficial influence of LSP on fatigue endurance has been reported in a number of recent works [e.g., 1-14]. A comprehensive review of recent achievements in this field has been provided by Wang, et al. [2].

In particular, an approximately 20 to 40 percent increase in fatigue strength of LSP’ed samples has been reported by Jin, et al. [3] and Zhang, et al. [4], respectively. In terms of fatigue life, the enhancement effect has been found to range from ~140 pct. [5] to ~300 pct. [6]. The beneficial influence of LSP may be further amplified using a double-sided treatment [7]. By this means, a ~170-pct. increase in fatigue endurance limit has been found in the work of Yang, et al. [8]. Moreover, the combination of LSP with conventional shot peening has been shown to result in a 9-fold increase in the number of cycles-to-failure [9]. As shown by Spanrad, et al. [10], Lin, et al. [11], and Zabeen, et al. [12], the superior fatigue properties of LSP’ed Ti-6Al-4V are associated with a retardation of both the nucleation and propagation of fatigue cracks by residual compressive stresses generated in near-surface areas. Furthermore, the distribution of residual stress (and thus fatigue resistance) was found to be highly sensitive to the particular LSP pattern/grid applied during the treatment [13]. A pattern consisting of multiple concentric rings of LSP spots was found to produce the most favorable distribution of residual stresses. Given the numerous benefits of LSP treatment, it has thus been recommended for use as a standard practice in damage-tolerant design [14]."

4- At the beginning of the materials and methods section, the impurity level of the consumed materials should be mentioned.

Authors’ response

According to the comment, the impurity content in the program material has been indicated in the revised manuscript (Page 2, Section 2, Paragraph 1):

"The material used in the present investigation was commercially-produced Ti-6Al-4V alloy in the mill-annealed condition. The nominal impurity content includes 0.25 wt. pct. (max) of iron and 0.2 wt. pct. (max) of oxygen."

Overall, authors are strongly advised to make the text coherent. The text and the structure of the article are not coherent. This will be boring for the readers.

According to the comment, the manuscript has been carefully revised in order to make it coherent.

Comments on the Quality of English Language

It must be rewritten according to the comments.

Authors’ response

The manuscript has been carefully revised according to the comments.

Reviewer 2 Report

    In this manuscript, the authors reported the plastic strain during laser shock (LSP) peening of the Ti-6Al-4V alloy though examining the microstructural response of the material by electron backscatter diffraction (EBSD) and nano-hardness measurements. The motivation of this work is well articulated in the introduction, followed by a detailed method and materials descriptions. In the result part, they indicated only minor changes in both the shape of α grains/particles and hardness due to a small plastic strain in the material during LSP peening. This is mainly attributed to the relatively-high strain hardening rate during LSP and the absence of dynamic recovery. The above results are supported by strong microstructural examination and analyses. This work contributes to fundamentally understanding the material behavior during the LSP treatment. The whole manuscript is also well organized. In my opinion, it can potentially be published in Materials after addressing the following several minor comments.

1) The arrows and ellipse in Fig. 3b should be mentioned in the figure caption. Besides, the figure captions of Figs. 4 (d) and 6(d) should be included. I guess that "(b-c)" in the figure captions should be "(b-d)".

2) There are two documents in the Supplementary Materials, in which the title "Evaluation of thermal effects induced during laser-shock peening" titled one should be consistent with that of the main manuscript if the document uploaded is correct. However, I guess that this probably be independent.

Author Response

Responses to the comments of Reviewer #2

 on the paper entitled “On the degree of plastic strain during laser shock peening of Ti-6Al-4V” (Manuscript ID: materials-2508720)

The authors would like to express their gratitude to Reviewer for his/her remarks. Below, we provided specific replies to the issues raised.

Note: Reviewer’s comments are highlighted with bold. 

In this manuscript, the authors reported the plastic strain during laser shock (LSP) peening of the Ti-6Al-4V alloy though examining the microstructural response of the material by electron backscatter diffraction (EBSD) and nano-hardness measurements. The motivation of this work is well articulated in the introduction, followed by a detailed method and materials descriptions. In the result part, they indicated only minor changes in both the shape of α grains/particles and hardness due to a small plastic strain in the material during LSP peening. This is mainly attributed to the relatively-high strain hardening rate during LSP and the absence of dynamic recovery. The above results are supported by strong microstructural examination and analyses. This work contributes to fundamentally understanding the material behavior during the LSP treatment. The whole manuscript is also well organized. In my opinion, it can potentially be published in Materials after addressing the following several minor comments.

1) The arrows and ellipse in Fig. 3b should be mentioned in the figure caption. Besides, the figure captions of Figs. 4 (d) and 6(d) should be included. I guess that "(b-c)" in the figure captions should be "(b-d)".

Authors’ response

The captions to Figures 3, 4, and 6 have been revised according to the comment

"Figure 3. (a) Image-quality map taken after 3 successive LSP pulses from the material beneath LSP crater with selected area shown at higher magnification in (b). The laser peened surface is at the top. For illustrative purpose, β phase in (a) is highlighted with red; in α phase, blue lines indicate {10-12} twin boundaries and black broken lines show {11-20} plane traces which are closest to the orientation of deformation bands. In (b), arrows indicate the presumed amorphous layer, while the circled area shows a Widmanstatten (or martensitic alpha) microstructure."

"Figure 4. (a) Kernel-average-misorientation (KAM) map with overlaid IQ map taken after 3 successive LSP pulses from the material beneath LSP crater, and (b-d), misorientation profiles measured along the arrows labelled in (a). The color code for KAM angles is shown in the bottom right corner of (a). The selected area in (a) shows the presumed Widmanstatten (or martensitic) alpha structure."

"Figure 6. (a) Kernel-average-misorientation (KAM) map taken after 5 LSP passes from the material beneath pulsed area, and (b-d), misorientation profiles measured along the arrows shown in (a). The color code for KAM angles is shown in the bottom right corner of (a)."

2) There are two documents in the Supplementary Materials, in which the title "Evaluation of thermal effects induced during laser-shock peening" titled one should be consistent with that of the main manuscript if the document uploaded is correct. However, I guess that this probably be independent.

Authors’ response

The calculations provided in these supplementary materials shows that the thermal effect associated with LSP was presumably small. In turn, this result supports the validity of the analysis of the LSP-induced stress given in Section 4.2.

In order to avoid confusion, the following remark has been added to the revised manuscript (Page 10, footnote):

"From theoretical considerations, it was determined that thermal effects induced during LSP were mainly confined within the sacrificial layer and were small in the bulk of the Ti-6Al-4V workpiece (see details in the supplementary Word file “Supplementary material_Evaluation of thermal effect of LSP”). Hence, thermal effects were neglected during evaluation of the LSP stress."

Reviewer 3 Report

Comment 1: Surface treatments like shot peening and laser shock peening can help inhibit crack initiation by creating compressive residual stresses at the surface, which enhances fatigue strengths and life expectancy of the material. This study aims to evaluate the degree of plastic strain of Ti-6Al-4V introduced during laser shock peening. To achieve this, the microstructural responses of the LSP’ed Ti-6Al-4V samples were examined using EBSD and microhardness tests. The subject matter is of interest to scholars engaged in this area of study. In my opinion, this study lacks sufficiently in-depth scope and conclusions that could potentially foster innovation and advancement within this field. Therefore, the reviewer suggest major revision before this paper may be considered for publication:

Comment 2: In the Introduction, the authors outline the “contradiction in experimental observations of LSP’ed samples” from previous studies. What are the possible reasons for that? How were their raw samples fabricated and what were their initial microstructure? Differences in initial microstructure could be one possible reason for the “contradictions”. Moreover, how would this study’s results contribute to the scientific debate?

Comment 3: How were the original mill-annealed samples prepared? Please provide the process parameters of the heat treatment.  

Comment 4: In Section 3.1.1, the following paragraph “In particular, the formation of a narrow dark-contrast layer at the crater surface is of interest (arrows in Figure 3b). High-resolution EBSD revealed no measurable variation in contrast (i.e., the presence of internal structure) within this layer, thus suggesting its amorphous nature. Indeed, amorphization of surface material during high-energy laser pulsing has been reported sometimes [28, 29]. Although the origin of this phenomenon is still not completely clear, a plausible explanation is the occurrence of rapid solidification of ablated material [29].”  was repeated.

Comment 5: Some sentences are missing periods, such as the last sentences of Section 3.1.1, Section 3.1.2, and Section 3.2, etc.

Comment 6: The authors stated that the LSP process applied in this study was expected to induce more severe plastic strain than typical LSP conditions. Please clarify in a logical way that how the results from this study are also application to the LSP’ed Ti-6Al-4V in general.

Comment 7: Why didn’t the authors use XRD to determine the stress/strain state of the LSP’ed samples?

Comment 8: The title of this paper is “On the degree of plastic strain during laser shock peening of 2 Ti-6Al-4V” and the authors stated that the purpose of this study was to “qualitatively” evaluate the plastic strain of LSP’ed Ti-6Al-4V. However, the experimental results and the subsequent analysis were only able to reveal that “the strains are very small”. My concern is that could these findings contribute to the scientific community? I would suggest that the authors should better clarify this study’s novelty and contribution of this study in the Introduction & Conclusion.

Comment 9: In the Introduction, the authors mentioned that the present work was to validate the simulation results of plastic strain. However, the simulation was neither discussed in later sections nor directly confirm/disproved by this study. Could the simulation help with the stress/strain analyze for LSP process? The authors might consult this reference for the analysis of laser-Ti6Al4V interaction:  Peng Jin, Qian Tang, Kun Li, Qixiang Feng, Zhihao Ren, Jun Song, Yunfei Nie, Shuai Ma, The relationship between the macro- and microstructure and the mechanical properties of selective-laser-melted Ti6Al4V samples under low energy inputs: Simulation and experiment, Optics & Laser Technology,

Comment 10: The estimation process for peak imposed pressure and the strain hardening theory are both interesting, but they are mostly based on theories and assumptions. For example, how was the assumed accumulated strain of 0.1 selected? Is there any way to verify them, to some extent?

Minor editing of English language required

Author Response

Responses to the comments of Reviewer #3

 on the paper entitled “On the degree of plastic strain during laser shock peening of Ti-6Al-4V” (Manuscript ID: materials-2508720)

The authors would like to express their gratitude to Reviewer for his/her remarks. Below, we provided specific replies to the issues raised.

Note: Reviewer’s comments are highlighted with bold. 

Comment 1: Surface treatments like shot peening and laser shock peening can help inhibit crack initiation by creating compressive residual stresses at the surface, which enhances fatigue strengths and life expectancy of the material. This study aims to evaluate the degree of plastic strain of Ti-6Al-4V introduced during laser shock peening. To achieve this, the microstructural responses of the LSP’ed Ti-6Al-4V samples were examined using EBSD and microhardness tests. The subject matter is of interest to scholars engaged in this area of study. In my opinion, this study lacks sufficiently in-depth scope and conclusions that could potentially foster innovation and advancement within this field. Therefore, the reviewer suggest major revision before this paper may be considered for publication:

Authors’ response

The manuscript has been carefully revised in accordance with the comments provided by Reviewer (given below). The authors do hope that the revised version of the manuscript is suitable for publication.

Comment 2: In the Introduction, the authors outline the “contradiction in experimental observations of LSP’ed samples” from previous studies. What are the possible reasons for that? How were their raw samples fabricated and what were their initial microstructure? Differences in initial microstructure could be one possible reason for the “contradictions”. Moreover, how would this study’s results contribute to the scientific debate?

Authors’ response

The survey of the available scientific literature revealed a contradiction regarding the degree of plastic deformation induced during laser shock peening (LSP). Specifically, a number of microstructural observations suggested that LSP resulted in a large (or even severe) plastic deformation (Refs. 3, 15-17, 21-24) while other works indicated that the LSP-induced strain is likely small.

Based on the microstructural observations of the present study, the authors suggested that the microstructural evidences of the large plastic strain (reported in Refs. 3, 15-17, 21-24) may be artifacts associated with a partial ablation of the surface material during LSP. One of the consequences of this phenomenon may be the development of the nano-scale Widmanstatten/martensitic-alpha microstructure, which is morphologically similar to the microstructure produced by the extensive mechanical twinning.

On the other hand, the authors do agree with Reviewer that the revealed contradiction may be due to the difference in the initial microstructures of program material.

Thus, to clarify this issue, the following remark has been added to the revised manuscript (Page 6, Paragraphs 2 to 3):      

"It is also worth noting that the microstructure within the circled area in Figure 3b exhibited what appears to be either a Widmanstatten or a martensitic alpha microstructure. In titanium alloys, such microstructures are usually associated with high cooling rates [e.g., 30, 31]. It is also interesting to note that the morphology of the Widmanstatten structure is somewhat similar to microstructural patterns sometimes ascribed to multiple twinning in several previous studies of LSP’ed Ti-6Al-4V (supplementary Figure S2).

From the above considerations, it is possible that the development of nanocrystalline structure as well as extensive multiple twinning, which are sometimes reported in the scientific literature [e.g. 3, 15-17], are actually artefacts associated with the rapid solidification of ablated material. On the other hand, it should be noted that the program material used in most prior efforts had a transformed-b microstructure [3, 15, 17], in contrast to the mill-annealed material employed in the present study. Hence, the possibility exists that differences in structural response to LSP was related to initial microstructure."

Comment 3: How were the original mill-annealed samples prepared? Please provide the process parameters of the heat treatment.  

Authors’ response

According to the comment, the appropriate details of the mill-annealing procedure have been added to the revised manuscript (Page 2, Section 2, Paragraph 1):

"...Standard mill-processing involves hot rolling at 900 oC followed by annealing at 700 oC and final air cooling…"

Comment 4: In Section 3.1.1, the following paragraph “In particular, the formation of a narrow dark-contrast layer at the crater surface is of interest (arrows in Figure 3b). High-resolution EBSD revealed no measurable variation in contrast (i.e., the presence of internal structure) within this layer, thus suggesting its amorphous nature. Indeed, amorphization of surface material during high-energy laser pulsing has been reported sometimes [28, 29]. Although the origin of this phenomenon is still not completely clear, a plausible explanation is the occurrence of rapid solidification of ablated material [29].”  was repeated.

Authors’ response

The repeated text has been removed from the manuscript, according to the comment.

Comment 5: Some sentences are missing periods, such as the last sentences of Section 3.1.1, Section 3.1.2, and Section 3.2, etc.

Authors’ response

According to the comment, the manuscript has been carefully checked with regard to English grammar usage.

Comment 6: The authors stated that the LSP process applied in this study was expected to induce more severe plastic strain than typical LSP conditions. Please clarify in a logical way that how the results from this study are also application to the LSP’ed Ti-6Al-4V in general.

Authors’ response

To clarify this issue, the following remark has been added to the revised manuscript (Page 10, Paragraphs 2 to 3):

"The entire set of experimental results thus suggested that the magnitude of the LSP-induced plastic strain was low…

In the context of the above conclusion, it is worth noting that the LSP conditions in the present study were selected to provide severe deformation (i.e., a power density of 25 GW/cm2 and 5 successive LSP pulses to nearly the same location). Nevertheless, the generated plastic strain was found to be low. Therefore, it is likely that a typical LSP condition, which involves much less severe deformation conditions, should result in an even smaller strain. Hence, the conclusions derived in this work are likely applicable to the LSP of Ti-6Al-4V in general."

Comment 7: Why didn’t the authors use XRD to determine the stress/strain state of the LSP’ed samples?

Authors’ response

According to the comment, the appropriate explanation has been added to the revised manuscript (Page 3, Paragraph 3):

"The LSP-induced microstructures were characterized using the electron backscatter diffraction (EBSD) technique. This choice was dictated by the extremely-localized character of microstructure changes during LSP. According to the scientific literature [18, 19], the plastically-deformed zone developed during LSP of Ti-6Al-4V may be only a few grains deep. Hence, the preparation of an appropriate microstructural sample for transmission-electron microscopy (TEM) or x-ray diffraction (XRD) analysis is technically challenging. An additional issue is the inhomogeneous distribution of microstructure within the processed zone. Hence, statistical reliability is of particular importance for its examination. By contrast, the advanced capabilities of EBSD provide in-depth insight into microstructure, thus enabling a more or less complete picture of underlying processes."

Comment 8: The title of this paper is “On the degree of plastic strain during laser shock peening of 2 Ti-6Al-4V” and the authors stated that the purpose of this study was to “qualitatively” evaluate the plastic strain of LSP’ed Ti-6Al-4V. However, the experimental results and the subsequent analysis were only able to reveal that “the strains are very small”. My concern is that could these findings contribute to the scientific community? I would suggest that the authors should better clarify this study’s novelty and contribution of this study in the Introduction & Conclusion.

Authors’ response

According to the comment, the novelty and contribution of the present study have been highlighted in Introduction and Conclusion sections of the revised manuscript.

Introduction (Page 2, Paragraphs 3 to 4):

"…A number of microstructural observations have suggested that LSP results in large (or even severe) plastic deformation [3, 15-17, 21-24] while others appear to indicate a small LSP-induced strain [18-20].

In an attempt to shed more light on the structural response of Ti-6Al-4V to LSP, the present work was undertaken with the purpose to evaluate the degree of the LSP-induced plastic strain (at least qualitatively)… To the best of the authors’ knowledge, the present work is one of the first systematic studies in this field. It is believed that elucidation of this issue should contribute to our fundamental understanding of material behavior under the extreme conditions of LSP."

Conclusions (Page 12, Paragraph 6):

"The present work was undertaken to fill the gap in the current scientific literature regarding the degree of plastic strain induced during LSP. It was believed that an elucidation of this issue would provide fundamental insight into behavior of structural materials under the extreme conditions of LSP."

Comment 9: In the Introduction, the authors mentioned that the present work was to validate the simulation results of plastic strain. However, the simulation was neither discussed in later sections nor directly confirm/disproved by this study. Could the simulation help with the stress/strain analyze for LSP process? The authors might consult this reference for the analysis of laser-Ti6Al4V interaction:  Peng Jin, Qian Tang, Kun Li, Qixiang Feng, Zhihao Ren, Jun Song, Yunfei Nie, Shuai Ma, The relationship between the macro- and microstructure and the mechanical properties of selective-laser-melted Ti6Al4V samples under low energy inputs: Simulation and experiment, Optics & Laser Technology,

Authors’ response

According to the comment, the following discussion has been added to the revised manuscript (Page 10, Paragraph 2):

"The entire set of experimental results thus suggested that the magnitude of the LSP-induced plastic strain was low. This conclusion is consistent with numerical simulations of LSP [27], according to which the true plastic strains generated during LSP of Ti-6Al-4V were as low as 0.01. In turn, the comparability of these results is perhaps indicative of the feasibility of numerical approaches for an analysis of the behavior of Ti-6Al-4V under laser treatment, as highlighted by Jin et al. [33]."

References (Page 15):

"[33] Jin, P.; Tang, Q.; Li, K.; Feng, Q.; Ren, Z.; Song, J.; Nie, Y.; Ma, S. The Relationship between the Macro- and Microstructure and the Mechanical Properties of Selective-Laser-Melted Ti6Al4V Samples under Low Energy Inputs: Simulation and Experiment. Optic. Laser Technol. 2022, 148, 107713. "  

Comment 10: The estimation process for peak imposed pressure and the strain hardening theory are both interesting, but they are mostly based on theories and assumptions. For example, how was the assumed accumulated strain of 0.1 selected? Is there any way to verify them, to some extent?

Authors’ response

The degree of the LSP induced strain has been suggested from the results of numerical simulation reported in Ref. [27]. To avoid confusion, this issue has been emphasized in the revised manuscript (Page 2, Paragraph 5): 

"…According to the recent numerical simulations, this strain could be as low as ~0.01 [27]…"

Regarding the experimental verification of this result, there is no direct approach for this purpose, to the best of the authors’ knowledge. To clarify this issue, the following remark has been added to the revised manuscript (Page 10, Paragraph 6):

"..it is important to emphasize that experimental measurements of dynamic strength (and strain) are limited by the highest strain rate achievable using the split Hopkinson bar tests, i.e., 104 s-1. This is two orders lower that the typical LSP strain rate…"

Comments on the Quality of English Language

Minor editing of English language required

Authors’ response

According to the comment, the manuscript has been carefully checked with regard to English grammar usage.

Round 2

Reviewer 1 Report

Now, the manuscript can be published.

Now, the manuscript can be published.

Reviewer 3 Report

The authors addressed all the comments, which can be recommended for publicatioin.